# ZnO Nanoflower-Based NanoPCR as an Efficient Diagnostic Tool for Quick Diagnosis of Canine Vector-Borne Pathogens

**DOI:** 10.3390/pathogens9020122

**Published:** 2020-02-14

**Authors:** Archana Upadhyay, Huan Yang, Bilal Zaman, Lei Zhang, Yundi Wu, Jinhua Wang, Jianguo Zhao, Chenghong Liao, Qian Han

**Affiliations:** 1Laboratory of Tropical Veterinary Medicine and Vector Biology, School of Life and Pharmaceutical Sciences, Hainan University, Haikou 570228, China; Archana@hainu.edu.cn (A.U.); zhanglei@hainu.edu.cn (L.Z.); liwangjinhua@163.com (J.W.); jianguolxx@hainu.edu.cn (J.Z.); 2State Key Laboratory of Marine Resource Utilization in South China Sea, College of Material Science and Engineering, Haikou 570228, China; huanhuanyang@hainu.edu.cn; 3State Key Laboratory of Marine Resource Utilization in South China Sea, Hainan Provincial Key Laboratory of Research on Utilization of Si-Zr-Ti Resources, College of Material Science and Engineering, Hainan University, Haikou 570228, China; 4State Key Laboratory of Marine Resource Utilization in South China Sea, Hainan University, Haikou 570228, China; wuyundi@hainanu.edu.cn

**Keywords:** ZnO nanoflowers, PCR, nanomaterial-assisted polymerase chain reaction, nanoPCR, canine vector-borne diseases, *Babesia canis vogeli*, *Hepatozoon canis*

## Abstract

Polymerase chain reaction (PCR) is a unique technique in molecular biology and biotechnology for amplifying target DNA strands, and is also considered as a gold standard for the diagnosis of many canine diseases as well as many other infectious diseases. However, PCR still faces many challenges and issues related to its sensitivity, specificity, efficiency, and turnaround time. To address these issues, we described the use of unique ZnO nanoflowers in PCR reaction and an efficient ZnO nanoflower-based PCR (nanoPCR) for the molecular diagnosis of canine vector-borne diseases (CVBDs). A total of 1 mM of an aqueous solution of ZnO nanoflowers incorporated in PCR showed a significant enhancement of the PCR assay with respect to its sensitivity and specificity for the diagnosis of two important CVBDs, *Babesia canis vogeli* and *Hepatozoon canis*. Interestingly, it drastically reduced the turnaround time of the PCR assay without compromising the yield of the amplified DNA, which can be of benefit for veterinary practitioners for the improved management of diseases. This can be attributed to the favorable adsorption of ZnO nanoflowers to the DNA and thermal conductivity of ZnO nanoflowers. The unique ZnO nanoflower-assisted nanoPCR greatly improved the yield, purity, and quality of the amplified products, but the mechanism behind these properties and the effects and changes due to the different concentrations of ZnO nanoflowers in the PCR system needs to be further studied.

## 1. Introduction

Polymerase chain reaction (PCR), which is one of the most reliable and popular diagnostic techniques, was invented by Kary Mullis back in 1985 [1]. It is a molecular diagnostic system that has become one of the most important technologies in biology and medicine. PCR technique has a broad array of applications, which include mutation detection [2], gene cloning [3], genotyping [4], microarray [5], DNA sequencing [6], fingerprinting [7], paternity testing [8], pathogen detection [9], forensics [10] and diagnostics [11]. The ability to amplify a low copy number of DNA feasibly has made in vitro PCR one of the most important techniques in molecular biology. However, PCR technology has its own set of shortcomings and drawbacks, due to which its reliability sometimes can be questionable and debatable. This can be due to certain obstacles and inhibitors such as nonspecific by-products, low yield, and complexity due to guanine and cytosine (GC)-rich nature. PCR inhibitors have a direct effect on the reaction due to their interactions with the nucleic acid and/or their interference with the DNA polymerases or other thermostable enzymes [12]. This type of interaction of binding between to the nucleic acid may alter amplification and can lead to the co-purification of inhibitor and DNA [13]. DNA polymerases can be directly attacked by the inhibitors to block or alter their enzyme activity due to their cofactor requirements [13]. Magnesium is a critical cofactor, and agents that reduce Mg^2+^ availability or interfere with binding of Mg^2+^ to the DNA polymerase can inhibit PCR [14]. Sensitivity and efficiency of the PCR techniques can be achieved to a certain extent, but not completely, by optimizing certain critical factors such as the magnesium ion concentration, annealing temperature, cycle numbers, template quality, the concentration of DNA polymerase enzyme and incorporation of various additives. Despite the various time-consuming optimization measures and steps, the efficiency of PCR still remains a problem and the enhancement of the PCR techniques becomes commanding with respect to the current and ongoing challenges in experimental and clinical biology. In order to exploit the PCR technique completely, all the above issues need to be addressed appropriately. Many scientists and researchers have often incorporated various chemical and biological additives such as glycerol [15], formamide [16], betaine [17], and Dimethyl sulfoxide (DMSO) [18] in PCR to overcome these obstacles. Newer PCR techniques such as hot start PCR [19] and touchdown PCR [19] were developed to gain better efficiency with specificity in the reaction.

In the last few years, nanotechnology has provided a breakthrough with effective solutions to many problems in various scientific fields including biotechnology and, therefore, the PCR techniques have been greatly privileged due to this “nano eon”. In the 1980s, since nanotechnology started to gain prominence [20], nanomaterials have become more popular in diverse disciplines due to their versatile and unique properties such as high thermal conductivity and high surface to volume ratios [21,22,23]. Currently, the application of nanomaterials in the biomedical fields is mainly focused on biochips [24], pharmacotherapy [25], nano-bioprobe [26], biosensor [27], biomedical detection, and diagnosis [28,29]. Nanomaterials have been widely used for the development of a variety of diagnostic techniques that are less time-consuming, and are more efficient and user-friendly due to their unique optical, magnetic, electrical, and thermal properties [30]. These features of nanomaterials definitely simplify diagnostic procedures. Such nanomaterials, especially nanoflowers, represent breakthrough developments in nanotechnology for various disease diagnostic systems. Nanomaterial-assisted PCR (nanoPCR) [31] is a technique that facilitates the incorporation of nanomaterials into PCR reaction to achieve greater specificity and efficiency. Initially, a few researchers started working on nanomaterial-assisted PCR (nanoPCR) [32,33,34], and then many other research groups began to study and understand the mechanisms and interaction between nanomaterials and biological systems, thus furthering the applicability of these nanomaterials in molecular diagnostics and related areas [35,36]. In some research reports, nanomaterials such as carbon nanopowder [37], nano alloys [38], and multiwall carbon nanotubes [39] were successfully used to enhance and increase the efficiency of PCR. Some studies addressed an important point as to whether the nanomaterials compromised the fidelity of DNA replication in the process of PCR [40,41]. Other research reports have successfully used this technique for the diagnosis and detection of bacterial aerosols [42], porcine bocavirus [43], pseudorabies virus [44] and porcine parvovirus [45]. There are many nanomaterials that have been employed in PCR reactions to check their effects on the efficiency and specificity of the reactions, a few of which include gold nanoparticles (AuNPs) [46], graphene oxide (GO) [47], reduced graphene oxide (rGO) [47], quantum dots (QDs) [48], upconversion nanoparticles (UCNPs) [49], fullerenes (C60) [50], carbon nanotubes (CNTs) [51], nanocomposites [51], and many more. However, zinc oxide (ZnO) has attained great popularity in the biomedical field in recent years [52,53,54,55]. ZnO nanomaterials, when compared with the traditionally used gold nanoparticles, exhibit better properties such as surface area, good biocompatibility, chemical stability, and electrochemical activity. Additionally, the synthesis of ZnO nanomaterials is convenient, easy, hassle-free, economical, and eco-friendly, and thus it proves to be more favorable in cases of clinical detection [56]. Some studies have demonstrated that ZnO can bind to the enzymes to enhance the enzyme activity [57,58,59,60]. Scientists and researchers have been interested in nanoflowers, a relatively newer type of nanostructure, due to the positional characteristics of nanolayers. These nanolayers have a special area, that facilitates a higher surface-to-volume ratio when compared to classic spherical nanoparticles, which in turn drastically shoot up the efficiency of surface reactions for nanoflowers [60]. Development of a ZnO nanoflower or flower-like ZnO-assisted nanoPCR for the efficient and quick diagnosis of canine vector-borne diseases (CVBDs) has rarely been reported [61].

In this study, ZnO nanoflowers were synthesized and incorporated in PCR assay for the detection of two important CVBDs, *Babesia canis vogeli* and *Hepatozoon canis*, from a dog DNA sample. The ZnO nanoflower-assisted nanoPCR not only showed a drastic improvement in the efficiency and yield, but also reduced the time of the PCR assay as well. Employment of ZnO nanoflowers showed a significant effect on the overall PCR performance and as an effective diagnostic test for CVBDs. 

## 2. Materials and Methods

### 2.1. ZnO Nanoflower Synthesis and Characterization

ZnO nanoflowers or flower-like ZnO nanomaterials were synthesized using hydrothermal protocols in the laboratory settings. ZnO nanoflowers were synthesized by dissolving 1.5 g of zinc acetate, Zn(CH_3_COO)_2_ H_2_O, in a mixture of a solution containing 20 mL and 10 mL of ethanol and of de-ionized water, respectively, under vigorous stirring. Ammonia water (NH_3_ H_2_O) was added dropwise into the above mixture until a pH of 10 was achieved. All this procedure took place under continuous magnetic stirring and at room temperature. This solution was then transferred or poured into a 50 mL Teflon-lined autoclave, which was further incubated at 140 °C for 10 h. After the incubation period was over, the white-colored product from the Teflon-lined autoclave was collected and then underwent washing steps with de-ionized water and absolute ethanol, followed by centrifugation. These products were further dried at 60 °C for 4 h, followed by annealing at 500 °C for 2 h in a high-temperature furnace.

The morphology and purity of the ZnO nanoflowers were characterized through X-ray diffraction (XRD, Bruker D8 Advance, Germany) using CuKa radiation (h = 0.15406 nm) at a rate of 5 °C/min and a scanning range of 10–90 °C. The surface micro-morphology was observed by using a scanning electron microscope (S-4800, Hitachi, Japan), which was operated at a voltage of 10 kV, sputter-coatedwith gold, installed in the College of Material Science and Engineering, Hainan University, Haikou, China.

### 2.2. Sample and DNA Isolation 

Previously collected and identified positive blood samples for *B. canis vogeli* and *H. canis* from dogs of Hainan province, which were confirmed by DNA sequencing and were processed for fresh DNA isolation (unpublished data). Total DNA was extracted from 200 μL of Ethylenediaminetetraacetic acid (EDTA) anti-coagulated blood samples by using TIANamp blood DNA kit (TIANGEN, China) and Sangon Biotech Ezup blood DNA kit, as per the manufacturer’s instructions. DNA concentrations and purities were determined by measuring the absorbance using a NanoDrop Spectrophotometer (Thermo Scientific, Waltham, MA, USA). The extracted DNA samples were eluted in nuclease-free water and processed and/or stored at − 20 °C for downstream applications.

### 2.3. ZnO Nanoflower-Assisted PCR

Originally, a set of previously identified samples (unpublished data), 12 positive samples for *Hepatozoon canis* and 5 positive samples of *Babesia canis*, were tested in order to determine the sensitivity of the assay with regards to ZnO nanoflowers. Additionally, the concentrations of the DNA before and after the assay for the samples were identified and recorded. Tests were performed in triplicate, as mentioned. Hence, we concluded that ZnO nanoflower-assisted PCR is much more efficient, sensitive, and better than the normal PCR, owing to its advantages over the latter. The isolated DNA samples of *B. canis vogeli* and *H. canis* were used as template DNA in PCR assay. Two sets of PCR assays, set A and set B, with each set containing the same *B. canis vogeli*- and *H. canis*-positive DNA samples as a template, were performed in order to study the effect of ZnO nanoflowers on the PCR technique. A total of 1 mM of an aqueous solution of the synthesized ZnO nanoflower stock solution (flower-like ZnO) was prepared using Diethyl pyrocarbonate(DEPC)-treated water, which was autoclaved for 25 min prior to use. Then, 5 μL of nanomaterial solution from the prepared ZnO aqueous solution was added in the PCR mix according to Table 1. The description of the sets A and B are shown in Table 1. The primers and their expected sizes used for detecting *B. canis vogeli* and *H. canis* from dog DNA are summarized in Table 2.

End-point PCR assays were performed for both the sets A and B, following the protocol described in Table 1 to form a total reaction volume of 25 μL, which consisted of 2 μL of DNA template. Negative (no DNA) controls were included for all the PCR tests. The reaction mixtures were cycled in an Eppendorf gradient thermal cycler (Eppendorf, Germany). PCR products were examined on 2% agarose gel stained with 0.4 μg/mL ethidium bromide using a Quick-Load 5 kb DNA Ladder marker (TAKARA BIO, Inc., Beijing, China), visualized under the Gel Doc XR^+^ imaging system (BIO-RAD Laboratories, Inc., Hercules, CA, USA). 

### 2.4. Concentration of the Amplified DNA

The concentrations and purities of the amplified DNA from both sets of PCR assays were determined by recording the absorbance and values using a NanoDrop Spectrophotometer (Thermo Scientific, USA).

### 2.5. Statistical Analysis

Statistix 8.1 software was used to calculate the least significant difference (LSD) and for the calculation of the *p*-value for checking whether the difference between the concentrations of the amplified DNA samples of the two PCR groups, set A and B, with one PCR set with the nanomaterials and the other without nanomaterials, was statistically significant or not. 

### 2.6. Ethical Statement

The care and use of dogs and samples in this study was approved by Hainan University Institutional Animal Care and Use Committee. The collection of blood from dogs was supervised by the veterinarians of Hainan University. This study did not involve any endangered species, and no specific approvals and permissions were required.

## 3. Results

### 3.1. Structure and Morphological Analysis of ZnO Nanoflowers

The structure and purity of the ZnO nanoflowers were determined and demonstrated by powder X-ray diffraction (XRD). The XRD patterns and diffraction peaks of the ZnO nanoflowers are shown in Figure 1. The diffraction peaks were in good conformity with the standard card of ZnO powder diffraction file (PDF) #36-1451. No dissimilar peaks referring to impurities were seen. As no other irrelavant peaks when compared to the standard card were observed, the synthesized nanoflowers were considered as pure ZnO. This gave a clear indication that the synthesized ZnO nanoflowers were highly pure. The size and micromorphology of the ZnO nanomaterials, as in ZnO nanoflowers, was examined in detail by using the advanced protocol of scanning electron microscopy (SEM). Scanning electron micrographs of the ZnO nanoflowers are shown in Figure 2. The SEM images showed that the synthesized ZnO nanoflowers were self-assembled and clearly potrayed the nanopetal-like structure arising from the center of the flowers (Figure 2b). The synthesized ZnO nanoflowers showed a clear, uncongested, and good dispersity, and had an average diameter of about 1–2 µm.

### 3.2. ZnO Nanoflower-Assisted NanoPCR Effects of ZnO Nanoflowers on PCR Assay

Amplification of DNA with their respective band sizes (approximately 619 bp for *B. canis vogeli* and 666 bp for *H. canis*) was successfully achieved in both the sets A and B, which were in compliance with previous reports [62,63]. 

**Set A—**This set consisted of two known positive samples: positive *B. canis vogeli* DNA with ZnO nanoflowers (B1) and positive *B. canis vogeli* DNA without ZnO nanoflowers (B2), and positive *H. canis* DNA without ZnO nanoflowers (H1) and positive *H. canis* DNA with ZnO nanoflowers (H2). PCR was carried out for all samples followed by agarose gel electrophoresis, revealing some interesting results, as shown in Figure 3a, where sample B1 showed a clearly brighter band as compared to sample B2. In Figure 3b, the sample H2 showed a clearly brighter band than H1. These results clearly suggested that after the addition of ZnO nanoflowers in the reaction, a visible difference in the appearance of the bands with respect to their brightness and sharpness was found. 

**Set B—**This set also consisted of two known positive samples: positive *B. canis vogeli* DNA with ZnO nanoflowers (B3) and positive *B. canis vogeli* DNA without ZnO nanoflowers (B4), and positive *H. canis* DNA with ZnO nanoflowers (H3) and positive *H. canis* DNA without ZnO nanoflowers (H4). In addition, the PCR for set B was carried out using the modified thermal cycling conditions as mentioned in Table 1 with 25 cycles. PCR was carried out for all samples, followed by agarose gel electrophoresis revealing some interesting results, as shown in Figure 4a, where sample B3 showed a clear, brighter, and visible band as compared to sample B4, which showed no specific band. In Figure 4b, the sample H3 showed a visibly clearer and brighter band than H4, which obviously showed no specific band. 

The obtained results indicated that after the addition of ZnO nanoflowers in the reaction there was a visible difference in the appearance of the bands with respect to their brightness and sharpness. Even after following the modified thermal cycling conditions and reducing the number of cycles to 25 cycles, we could still obtain sharp, clear, and bright bands after the addition of ZnO nanoflowers. This reduced the reaction time by almost 45–50%. 

### 3.3. Concentration, Yield, and Purity of the Amplified DNA

Significant improvement in the DNA yield, concentration, and purity were observed post-amplification. Incorporation of the ZnO nanomaterials–ZnO nanoflowers in the PCR reaction led to a better yield and concentration of the amplified DNA. The concentration was measured by using NanoDrop. A large amount of DNA template may give rise to several non-specific PCR products. On the other hand, lower amounts of DNA may reduce the efficacy and accuracy of the PCR amplification. On measuring the concentrations of the amplified DNA, it was found that samples B1 and H1 (without ZnO nanoflowers) had a significantly lower concentration of DNA than the samples B2 and H2 (with ZnO nanoflowers), as shown in Table 3. Moreover, samples subjected to modified thermal cycling conditions (B3, H3, B4, and H4) demonstrated interesting results, also shown in Table 3. The results obtained were consistent and reproducible.

Most importantly, the overall PCR assay time was substantially reduced by 1 h 15 min by using the modified thermal cycling conditions, without compromising on the concentration, yield, and efficiency of the reaction and DNA. Our results clearly suggested that the entire PCR assay was completed in just 1 h and 10 min, whereas the normal PCR takes around 2 hut and 20 min to complete. The cycle number was reduced to 25 cycles coupled with the incorporation of ZnO nanoflowers in the PCR system. It was clear from the obtained results that ZnO flower-based nanoPCR with modified thermal cycling conditions is a less time consuming and efficient PCR platform for the detection of CVBDs, without compromising on the quality of DNA. Hence, ZnO nanoflower-based nanoPCR can be of great benefit to the field of veterinary diagnostics. 

### 3.4. Statistical Analysis

The least significant difference (LSD) was calculated for the concentration of the amplified DNA of the two sets A and B by comparing the means of both sets using the Statistix 8.1 software. The current findings showed the difference between the DNA samples of two groups—PCR sets A and B—one with the ZnO nanoflowers and other without the ZnO nanoflowers, were statistically significant at *p* < 0.01 (Table 3).

## 4. Discussion

This study aimed to investigate the effects on the PCR conditions and their results before and after the addition of ZnO nanoflowers as a nanomaterial. The addition of 1 mM concentration of ZnO nanoflowers in the PCR system gave excellent results; however, in-depth and detailed study about the function, concentration, and mechanism of action of ZnO nanoflowers in PCR assay need to be further investigated. This kind of investigation might give rise to a newer, more rapid, and more efficient diagnostic system in veterinary diagnostics and the study of infectious diseases. An important property of ZnO nanoflowers, such as their good adsorption to DNA, improves the sensitivity and overall turnaround time of the assay. Because ZnO nanomaterials are said to have good compatibility with DNA molecules, they have the ability to protect DNA from damage [64]. 

Nanomaterials have gained a lot of popularity as positive and active modulators of PCR due to their heat transfer properties. Many researchers have worked with gold nanoparticles and have found them to be great influencing factors when it comes to the specificity and sensitivity of the PCR test [65]. Other derivatives of ZnO nanomaterials such as the tetrapod-like ZnO nanoparticles, single-walled carbon nanotube, and carbon nanopowder were found to have a positive influence on the efficiency and specificity of the PCR assay [30]. The first research study of nanoPCR mainly focused on PCR sensitivity, and the authors concluded that it was easy to get good quality amplification products after the addition of the appropriate amount of a nanomaterial [66]. It is possible to increase the efficiency of PCR without compromising on the yield of the amplification product in a shorter period of time [65]. The effect of gold nanomaterials on PCR has been attributed to their physical property or heat transfer efficiency, and it has been concluded that the heat conductivity of gold nanoparticles could be the reason for the improved efficiency of PCR assay and its lower time consumption [33]. The same point was noticed in another nanomaterial-assisted study in a silver nanomaterial-based PCR experiment [67]. PCR is considered as a gold standard in molecular diagnostics. Diverse additives and nanomaterials have greatly assisted in the evolution of PCR. Fortunately, with the ongoing research on nanomaterial-assisted PCR, it has become convenient to have a PCR assay that is more sensitive, more efficient, and more specific. It is worth mentioning that Au nanoparticles can retain higher specificity even at lower annealing temperatures. Moreover, gold nanomaterial-assisted PCR has the same effects as the classical hot-start PCR, which can provide excellent applicability in numerous biomedical studies [67]. 

ZnO and TiO_2_ nanoparticles not only provide thermal efficiency in a PCR reaction but also may contribute to the reduction of the PCR reaction time [68]. Hence, the nanoPCR assay does not only offer better results in less time, but this reduction in time of reaction can also reduce the running time of expensive thermal cyclers that consume a lot of electricity. On the basis of the usage of titanium dioxide (TiO_2_) nanoparticles and ZnO nanostructures, a sevenfold improvement in the PCR effieciency was observed specifically with TiO_2_ nanoparticles [69]. In a similar kind of experiment, a research group used 0.6 nM TiO_2_ as an optimized concentration in their PCR detection of bacterial aerosols [68]. Amine- and silica-functionalized ZnO tetrapods have also been incorporated in the PCR assay [70], in which the amine-functionalized ZnO tetrapods demonstrated a greater efficiency in PCR when compared to silica-functionalized tetrapods. Platinum (b-cyclodextrin capped)-assisted PCR did not affect the PCR efficiency and specificity, but it helped greatly in sensitivity and heat transfer, ultimately leading to the time-saving PCR assay [71]. 

The wide range of applications of nanoPCR and nanodiagnostics have great potential in the direction of veterinary diagnostics, especially in point-of-care testing, although there are many hurdles for field-based applications due to the employability of sophisticated thermal cyclers and skilled labor to perform the tests, which makes it limited to laboratory settings. On the other hand, many nanodiagnostic techniques are still in the preclinical stage, which suggests the need for clinical validation using a large set of samples to validate and standardize these diagnostic tools [72]. However, the usage and application of the ZnO nanoflower-assisted nanoPCR is not very deep in this research, yet it definitely provides a scope for the unique nanoflower-assisted veterinary diagnostic technologies. Additionally, the synthesis of ZnO nanoflowers is an easy and eco-frienldy process that can be carried out at any regular laboratory set-up. The ZnO nanoflowers were prepared by hydrothermal reaction using easily available chemicals in a laboratory set-up [61]. Owing to the fact that many laboratories do not have access to characterization methods used to characterize and identify the purity of ZnO nanoflowers, these specific nanoflowers can also be procured commercially, and the required concentration can be directly used in the PCR reaction. Alternatively, the ZnO nanoflowers can be made to order commercially, in accordance with the desired parameters with respect to the specific experiments, outlying the need for further tests to characterize the ZnO nanoflowers in resource-limited settings. ZnO nanoflowers-assisted PCR assay is a dynamic process involving many thermal cycling conditions, and it is hence very important to understand the relationship and the mechanism between the nanoflowers and the PCR components and system. Further research related to the mechanism and synthesis of economical, user-friendly, and environmentally friendly nanomaterials such as ZnO nanoflowers should be encouraged so as to effectively exploit nanotechnology in the fields of molecular diagnostics. These kinds of research studies can be further elaborated, investigated, and discussed, as such techniques can be a benefit for the field of veterinary diagnostics, where precision and turnaround time of the reaction and results play a critical role.

## 5. Conclusions

In the present study, we described the use of the unique ZnO nanoflowers in PCR reaction and ZnO nanoflower-based PCR (nanoPCR) for the efficient, less time-consuming, and molecular-based diagnosis of canine vector-borne diseases (CVBDs). On the basis of our findings, it was concluded that the ZnO nanoflower-assisted nanoPCR is much better than the normal PCR. Differences in the concentrations of the amplified DNA of both the sets A and B were statistically significant. ZnO nanoflowers are easy, hassle-free, and economical to synthesize, and these nanoflowers coupled with PCR (nanoPCR) have a wide range of application, not only in the early management in veterinary diagnostics, but also in the diagnosis of other infectious diseases. Further studies are recommended to elaborate, investigate, and discuss such techniques in detail, as they have significant potential in the field of veterinary diagnostics, where precision and turnaround time of the reaction and results play a critical role.

## Figures and Tables

**Figure 1 pathogens-09-00122-f001:**
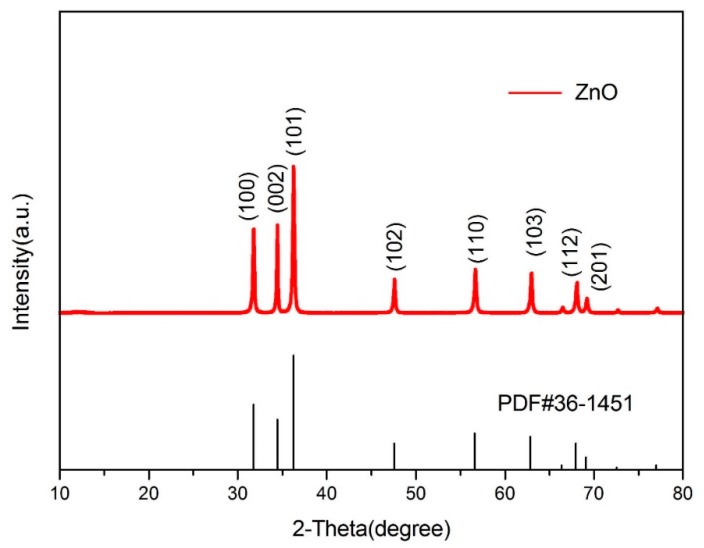
XRD patterns of ZnO nanoflowers when compared to the standard card of ZnO Powder diffraction file (PDF)#36-1451 showing exact similarity to the standard card patterns.

**Figure 2 pathogens-09-00122-f002:**
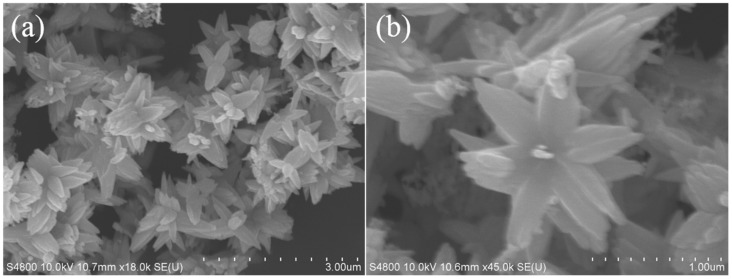
Scanning electron micrographs of ZnO: (**a**) low magnification with a diameter of 3.00 µm, (**b**) high magnification with a diameter of 1.00 µm.

**Figure 3 pathogens-09-00122-f003:**
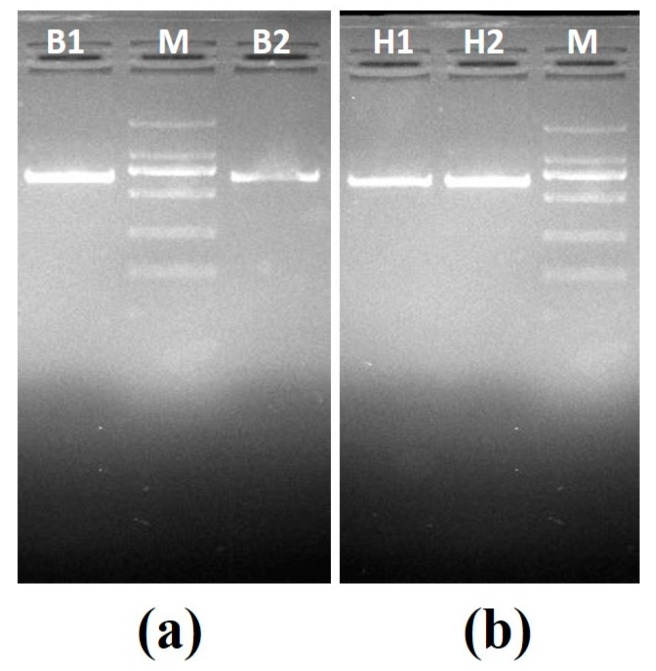
Agarose gel electrophoresis images generated by PCR set A: (**a**) agarose gel electrophoresis for *B. canis vogeli* DNA (619bp), B1—sample with ZnO nanoflowers, M—2000kb DNA marker, B2—sample without ZnO nanoflowers; (**b**) agarose gel electrophoresis for *H. canis* DNA (666bp), H1—sample without the ZnO nanoflowers, M—2000kb DNA marker, H2—sample with ZnO nanoflowers.

**Figure 4 pathogens-09-00122-f004:**
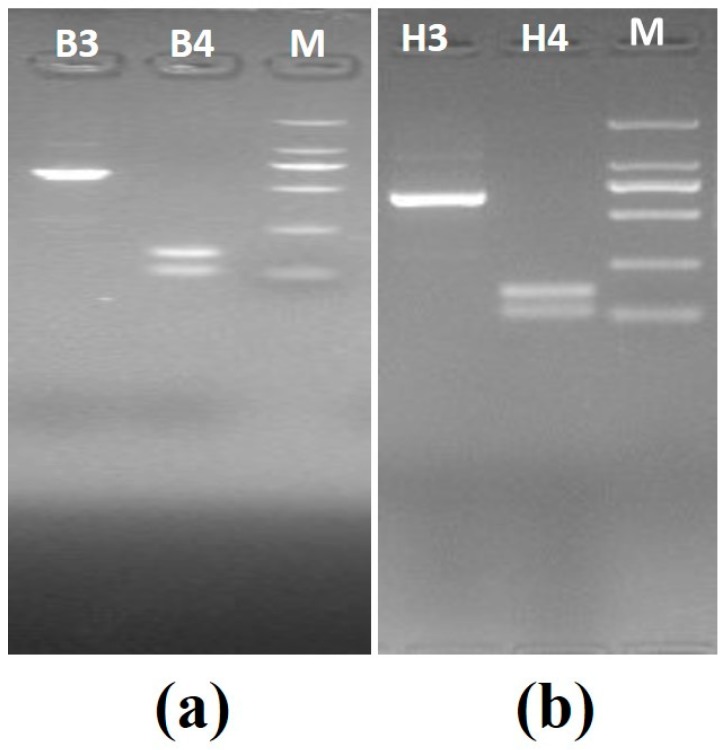
Agarose gel electrophoresis images generated by PCR set B (modified thermal cycling conditions): (**a**) agarose gel electrophoresis for *B. canis vogeli* DNA(619bp), B3—sample with the ZnO nanoflowers, M—2000kb DNA marker, B4—sample without ZnO nanoflowers; (**b**) agarose gel electrophoresis for *H. canis* DNA (666bp), H3—sample with the ZnO nanoflowers, M—2000kb DNA marker, H4—sample without ZnO nanoflowers.

**Table 1 pathogens-09-00122-t001:** PCR mix and thermal cycling conditions for sets A and B for the ZnO nanoflower-assisted nanomaterial-assisted PCR (nanoPCR).

**Set A**	**PCR Mix Components**	**PCR Mix 1 (without ZnO Nanoflower Solution)**	**PCR Mix 2 (with ZnO Nanoflower Solution)**	**Thermal Cycling Conditions for PCR**	
	PCR buffer mix (TransGen Biotech)	12.5 μL	12.5 μL	94 °C	5 min
	Primer (forward/reverse)	0.5 μL	0.5 μL	94 °C 54 °C 72 °C	1 min (35cycles)
	ddH_2_O	9.5 μL	4.5 μL	72 °C	5 min
	Nanomaterial (ZnO nanoflower solution)	NA	5 μL	4 °C	∞
	DNA (*Babesia canis vogeli/Hepatozoon canis*)	2 μL	2 μL		
**Set B**	**PCR Mix Components**	**PCR Mix 1 (without ZnO Nanoflower Solution)**	**PCR Mix** **2 (with ZnO Nanoflower Solution)**	**Thermal Cycling Conditions for PCR (Modified Conditions)**	
	PCR buffer mix (TransGen Biotech)	12.5 μL	12.5 μL	94 °C	2.5 min
	Primer (forward/reverse)	0.5 μL	0.5 μL	94 °C 54 °C 72 °C	30 s 1 min (25 cycles) 30 s
	ddH_2_O	9.5 μL	4.5 μL	72 °C	3 min
	Nanomaterial (ZnO nanoflower solution)	NA	5 μL	4 °C	∞
	DNA (*B. canis vogeli/H. canis*)	2 μL	2 μL		

**Table 2 pathogens-09-00122-t002:** Primer sets with respective product size for DNA amplification of the pathogens in dogs.

**Pathogen**	**Primer Sets**	**Product Size (bp)**	**Reference**
*B. canis vogeli*	Ba103F: CCAATCCTGACACAGGGAGGTAGTGACA Ba721R: CCCCAGAACCCAAAGACTTTGATTTCTCTCAAG	619	Kledmanee et al. (2009) [62]
*H. canis*	HEP-F: ATACATGAGCAAAATCTCAAC HEP-R: CTTATTATTCCATGCTGCAG	666	Inokuma et al. (2002) [63]

**Table 3 pathogens-09-00122-t003:** Concentrations and purity of the amplified DNA from sets A and B with statistical analysis.

**Set A**	**Sample**	**Average Concentration of Amplified DNA(ng/μL)** **± Standard Deviation**	**Purity** **A260/280**
	B1	676.2 ± 10 b	1.8
	B2	811.2 ± 12 a	1.8
	H1	799.5 ± 9 b	1.7
	H2	849.6 ± 8 a	1.8
**Set B**	**Sample**	**Concentration of Amplified DNA(ng/μL)** **± Standard Deviation**	**Purity** **A260/280**
	B3	550.2 ± 11 b	1.2
	B4	815.9 ± 10 a	1.8
	H3	648.6 ± 12 b	1.6
	H4	847.6 ± 9 a	1.8

*p* < 0.01.

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
