# Peer review of "ZnO Nanoflower-Based NanoPCR as an Efficient Diagnostic Tool for Quick Diagnosis of Canine Vector-Borne Pathogens"

_pathogens, 2020, doi:10.3390/pathogens9020122_

Round 1

Reviewer 1 Report

Archana et al. designed the study and well-performed. However, discussion and abstract should be written more precisely and focused. More detailed comments are below.

 English should be checked thoroughly throughout the manuscript. Tables should be placed in SI

Nanoflowers characterization should be supported by some more analysis such as TEM etc.

Reviewer 2 Report

The authors have produced a ZnO Nanoflowers to use it in a NanoPCR and to allow a higher sensitivity of the detection of two classical PCRs targeting Babesia canis vogeli and Hepatozoon canis.

The manuscript is well written, nevertheless major improvements are needed before publication. Indeed sensitivity and specificity of the PCR reactions are not well tested with and without the ZnO nanoflowers.

1/ In the introduction, reference start line 36 at the number 3 instead of number 1. The puiblication number 1 appears only line 94.

2/ In the introduction, line 43-47. Add also the problem of presence of inhibitors into the PCR reactions as an obstacle.

3/ Add a reference line 87

4/ Add a reference line 95

5/ Is it not clear if the same positive samples for Babesia canis and Hepatozoon canis are used in both sets A and B. Please clarify into the Material and methods part. Because of you statistical analysis performed, i think they are probably the same.

6/ My major concern is  linked with the conclusion provided by the authors regarding the higher sensitivity of the PCR reaction when they add the ZnO nanoflowers. The authors have only tested one positive sample for each pathogens, into two different set of PCR reaction with and without the ZnO nanoflowers. Even if the sensitivity was higher with the ZnO and more rapid, this was done only one time and with only one DNA of each pathogens. To confirm their findings, the authors should have tested different DNA of positives samples and dilution series to test sensitivity.

7/To allow the use of this ZnO nanoflowers in routine by other lab, this kind of nanoproduct should be availble directly into PCR mix or should be sell by a company. Only few laboratory will be able to produce their own ZnO nanflowers. This fact should be mention into the discussion section.

Round 2

Reviewer 2 Report

Minor revision: 

This statement (answer made by the authors to my first review) should be included into the material and method section to better explain the result obtain and the conclusion provided into the manuscript

"Originally, a set of previously identified (unpublished data), 12 positive
samples for Hepatozoon canis and 5 positive samples of Babesia canis were tested in order to determine the sensitivity of the assay with regards to ZnO nanoflowers. Additionally, the concentrations of the DNA before and after the assay for the samples were identified and recorded. Tests were done in triplicates as mentioned. Hence, we conclude that ZnO nanoflowers assisted
PCR is much efficient, sensitive and better than the normal PCR owing to its advantages over the latter."

Round 3

Reviewer 2 Report

No more comments